

# Relative Humidity Effect on the Formation of Highly Oxidized Molecules and
# New Particles during Monoterpene Oxidation
Xiaoxiao Li[1,2], Sabrina Chee[1], Jiming Hao[2], Jonathan P. D. Abbatt[3], Jingkun Jiang[2*], and James N.
Smith[1*]
[1]Chemistry Department, University of California, Irvine, CA 92697, USA
[2]State Key Joint Laboratory of Environment Simulation and Pollution Control, School of Environment, Tsinghua University,
Beijing, 100084, China
[3]Department of Chemistry, University of Toronto, Toronto, Canada
*: *Correspondence to*: J. N. Smith (jimsmith@uci.edu) and J. Jiang (jiangjk@tsinghua.edu.cn)
**Abstract.** It has been widely observed around the world that the frequency and intensity of new particle formation (NPF)
events are reduced during periods of high relative humidity (RH). The current study focuses on how RH affects the formation
of highly oxidized molecules (HOMs), which are key components of NPF and initial growth caused by oxidized organics. The
ozonolysis of α-pinene, limonene, and △³-carene, with and without OH-scavenger, were carried out under low NOx conditions
under a range of RH (from ~3% to ~90%) in a temperature-controlled flow tube. A Scanning Mobility Particle Sizer (SMPS)
was used to measure the size distribution of generated particles and a novel transverse-ionization chemical ionization inlet with
a high-resolution time-of-fight mass spectrometer detected HOMs. A major finding from this work is that neither the detected
HOMs nor their abundance changed significantly with RH, which indicates that the detected HOMs must be formed from
water-independent pathways. In fact, the distinguished OH- and $O_3$-derived peroxy radicals ($RO_2$), HOM monomers, and
HOM dimers could mostly be explained by the autoxidation of $RO_2$ followed by bimolecular reactions with other $RO_2$ or
hydroperoxy radicals ($HO_2$), rather than from a water-influenced pathway like through the formation of a stabilized Criegee
intermediate (sCI). However, as RH changed from 3 to 90% the particle number concentrations decreased by a factor of 2~3
while particle mass concentrations increased or decreased slightly within a factor of 2. These observations show that, while
high RH appears to inhibit NPF as evident by the decreasing number concentration, this reduction is not caused by a decrease
in $RO_2$-derived HOMs formation. One possible explanation is the existence of other extremely low volatility compounds
(ELVOCs), like gas phase formed sCI-included accretion products, which are responsible for the very first steps of NPF but are
not detected by nitrate-based chemical ionization mass spectrometry. These ELVOCs may be preferentially reduced at high
RH compared to more volatile compounds, the latter of which mainly determine the final mass concentration of particles.
Another possibility is that a fraction of HOMs cluster with water (but detected as the declustered molecules) at high RH in such
a way that they may no longer be able to participate in cluster formation, thereby suppressing NPF.



## 1 Introduction

New particle formation (NPF) is ubiquitous around the world (Kulmala et al., 2004). Newly formed particles contribute
greatly to global particle populations and can grow further to act as cloud condensation nuclei (CCN), thereby influencing
clouds and climate (Makkonen et al., 2012; Merikanto et al., 2009; Dunne et al., 2016). NPF characteristics vary from site to
site because of varying precursors and atmospheric conditions. It has been widely observed that the intensity (Sihto et al.,
2006; Dada et al., 2017) and frequency (Dada et al., 2017; Boy and Kulmala, 2002; Hyvönen et al., 2005) of continental NPF
are reduced during periods of high RH, resulting in reduced ultrafine particle number concentrations during these periods
(Weber et al., 1997). For example, 20 years of observations in the boreal forest at Hyytiälä, Finland, showed that NPF is
more likely during periods of low ambient RH (Dada et al., 2017). In central Amazonia, where particle composition is
dominated by oxidation products of biogenic organic compounds, new particles were not formed at ground-level where RH
was always higher than 60%, but rather formed in the upper troposphere where RH and condensation sink (CS) were
significantly lower (Poschl et al., 2010; Andreae et al., 2018). In urban areas, NPF also favors low RH (Cai et al., 2017; Shen
et al., 2011). Despite the low continental NPF event frequency at high RH, NPF has still been observed in the free
troposphere in vicinity of clouds, where RH is extremely high (Weber et al., 1999) and in coastal and marine areas where RH
is typically greater than 90% (O'Dowd et al., 1998).

The widely observed anti-correlation between NPF and RH in the field experiments can be attributed to the indirect
influence of water. For example, high RH often corresponds to greater cloud cover, which can lead to lower ground-level
concentrations of photo-oxidized precursors such as $H_2SO_4$ and highly oxidized molecules (HOMs) as well as an increased
condensation sink that leads to scavenging of precursors and clusters (Hamed et al., 2011). On the other hand, water vapor
may also directly influence NPF by regulating the formation of gas phase precursors or by participating in cluster formation.
For example, chamber and model experiments on the binary sulfuric acid-water system have demonstrated positive
relationships between particle formation rate and RH (Duplissy et al., 2016; Merikanto et al., 2016). While in the ternary
($H_2SO_4$/MSA-$H_2O$-Amine/$NH_3$) system, $H_2O$ was reported to have either positive (Chen et al., 2015) or negative (Napari et
al., 2002) effects on NPF. Some studies have hypothesized that high water content might suppress the formation of
NPF-related organics from the oxidation of biogenic precursors (Hyvönen et al., 2005; Boy and Kulmala, 2002). However,
no direct evidence of this has been provided.

Although sulfuric acid has been recognized as the most important precursor of new particle formation, it alone can't explain



the rapid formation and growth rates observed in the field (Kuang et al., 2008). Organic compounds, ammonia, amines, and
water are also likely involved (Zhang et al., 2012; Chen et al., 2012). Organics have been shown to be very important for
cluster formation and stabilization in theoretical studies (Ortega et al., 2016; Donahue et al., 2013), laboratory experiments
(Tröstl et al., 2016; Schobesberger et al., 2013) and field measurements (Bianchi et al., 2016; Hoffmann et al., 2001; Metzger
et al., 2010). Organics can either form clusters with sulfuric acid or purely with themselves (Zhao et al., 2013; Zhao et al.,
2009). They can also contribute significantly to the condensational growth of newly formed particles, determining particle
growth rates, particle lifetime, and global particle and CCN concentrations (Donahue et al., 2011; Vehkamäki and Riipinen,
2012). The ability of organics to take part in particle formation and condensational growth depends on their volatility as well
as reactivity. HOMs, such as extremely low volatility organic compounds (ELVOCs, saturation mass concentration ($C^*$) <
$3 \times 10^{-4}$ μg m$^{-3}$) or low volatility organic compounds (LVOCs, $3 \times 10^{-4}$ < $C^*$ < 0.3 μg m$^{-3}$), are likely contributors to NPF
(Donahue et al., 2012; Ehn et al., 2014).

Despite its large contribution to NPF, the direct measurement of HOMs has long been a challenge because of their low
atmospheric concentrations, low volatilities, and short lifetimes. Recently, the development of the high resolution
time-of-flight chemical ionization mass spectrometer (HRToF-CIMS) overcame this barrier and made the measurement and
identification of HOMs feasible (Junninen et al., 2010; Jokinen et al., 2012). HOMs from both monoterpene and aromatic
oxidation showed high O/C ratios of > 0.7-0.8, and were present as monomers, dimers and even higher order clusters
(Molteni et al., 2018; Ehn et al., 2012). These high O/C ratios could not be explained by any of the formerly known
oxidization pathways unless the autoxidation of RO$_2$ was taken into consideration (Crounse et al., 2013; Barsanti et al.,
2017). Autoxidation was widely observed in condensed phase reactions, however, it was not considered in the gas phase
previously because of the high energy barrier. This was further confirmed by the fact that at higher temperatures, more
HOMs are formed than at low temperatures (Frege et al., 2018).

Since most laboratory experiments related to the formation of HOMs have been conducted under conditions of constant RH,
usually low or medium RH of less than 60% (Ehn et al., 2012; Zhang et al., 2015), it was still unknown whether and how
water vapor might impact HOMs formation. High RH conditions are difficult to achieve in chamber experiments without
significantly changing temperature and pressure. In addition, HOMs detection by the current commercially available CIMS
inlet based on the design of Eisele and Tanner is subject to water cluster influence (Kuerten et al., 2016).

In this research, three different endocyclic monoterpenes, α-pinene, limonene and △³-carene were reacted with ozone, with





and without hydroxyl radical (OH) scavengers, in a reaction flow tube. RH influences on HOMs formation and
organic-driven NPF were studied under a range of RH from 0% to 90%. Generated closed-shell HOMs and $RO_2$ were
measured using a home-built CIMS inlet coupled to a HRToF mass spectrometer (LTOF mass analyzer, Tofwerk AG). The
CIMS inlet effectively reduced water clustering onto ions sampled into vacuum, thus removing sample artifacts caused by
high water vapor levels. Water vapor influence on the formation of $RO_2$, HOM monomers and HOM dimers were studied.
The volatility of $O_3$- and OH-derived closed-shell HOMs were estimated with a group contribution-based model (SIMPOL)
and a recently developed statistical model to study the potential contribution of $O_3$ and OH initiated chemistry on NPF.

## 2 Methods

### 2.1 Flow tube reactor

The experiments were performed in a laminar flow tube reactor consisting of a 150 cm long Pyrex glass cylinder with a
volume of 8.5 $dm^3$ (Figure 1). The flow tube was located in a temperature controlled room (T=293±2°C) and was covered so
that all experiments were performed under dark conditions. The monoterpenes were injected into the chamber using a
syringe pump (model NE-300, New Era Pump Systems, Inc.) evaporated into a 2.5 LPM flow of dry, purified "zero air." $O_3$
was generated by passing 0.5 LPM dry zero air (79% $N_2$, 21% $O_2$) over a Hg UV lamp (model 90-0004-04, UVP, LLC) and
then diluted with 6.5 LPM of zero air at the prescribed RH. The zero air was generated with a zero air generator (model
747-30, Aadco Instruments), with NOx and $SO_2$ concentrations specified to be less than 0.5 ppbv. Low NOx conditions were
achieved using zero air as-is. A temperature-controlled bubbler filled with deionized water was used to generate humid air,
and the prescribed RH was achieved by controlling the temperature of the bubbler. Gas inlets to the flow tube were made
from Teflon tubing that were capped and drilled with small holes to distribute sample air uniformly in the flow tube cross
section. The uniform distributions of $O_3$ and $H_2O$ in the flow tube were confirmed by measuring [$O_3$] and RH at the different
locations prior to the experiments. In every experiment, RH was adjusted to be constant for at least 30 min for each of the
four RH steps (0~5%, 30~38%, 58~65%, 85~92%). The total flow rate was kept at 8.5 LPM so that the average reaction time
was constant (~60s) for each experiment. At the beginning of the experiments, the inner wall of the reactor was washed with
ultra-pure water. All of the flow rates were calibrated before and during the experiments.

### 2.2 Instrumentation

#### 2.2.1 Transverse Ionization – Chemical Ionization Mass Spectrometer

A self-designed and home-built chemical ionization inlet, called Transverse Ionization (TI) inlet (Figure 2 and Figure S1),
was used in front of the LTOF mass analyzer. The TI design is similar to those of the Ambient-pressure Proton transfer Mass
Spectrometer (AmPMS) (Hanson et al., 2011) and the cluster-CIMS (Zhao et al., 2010). In the TI inlet, a 4-10 LPM flow of



sample air is passed across the inlet orifice of the mass spectrometer, where it encounters an orthogonal, 1 LPM reagent ion
gas flow consisting $N_2$ containing ionized nitrate ions ($NO_3^-$) as well as potential cluster ions $(HNO_3)_nNO_3$ with n=1-3. For
the current study, the sample flow to the inlet was set to 4.5 LPM. Chemical ionization occurs at atmospheric pressure and
temperature. The reagent gas is generated by passing 3 ccm of $N_2$ over a small vial containing nitric acid, which is then
ionized by a 370 MBq $Po^{210}$ radioactive source (model P-2021, NRD, LLC). An additional flow of $N_2$ can be added to the
reagent gas to change the reagent ion concentration, and the assembly can be adjusted to vary ion-molecule reaction time.
The latter can be controlled by adjusting the sample and reagent gas flow rates or by applying different voltages to the
ionization source and the main inlet block. To minimize the diffusion loss in sample lines, the inlet of the TI source was
connected to the flow tube outlet by a short (~10 cm) piece of electro-polished stainless steel tubing. Compared to the widely
used commercial nitrate source patterned after the design by Eisele and Tanner (1993) and marketed by Aerodyne, Inc., no
additional sheath flow is required so thus any impurities potentially introduced by the sheath flow are eliminated. Some flow
disturbance may occur where the sample flow encounters the transverse reagent flow, which may lead to non-ideal behavior.
However, even at the maximum total flow of 11 LPM, the Reynolds number in this region is ~500 and thus turbulence is not
expected to be significant.

Another unique aspect of the TI design is the use of an $N_2$ curtain gas in front of the inlet orifice to the mass spectrometer to
reduce water clustering on reagent and sample ions. Water clusters are expected to form at high RH mainly during the
free-jet expansion of the sampled gas on the vacuum side of the orifice plate (Thomson and Iribarne, 1979). The presence of
these clusters makes the identification and quantification of both sample and reagent ions challenging (Kulmala et al., 2014;
Lee et al., 2014; Kuerten et al., 2016; Ehn et al., 2014). Figure 2 shows the details of the TI source that address this issue.
Small holes drilled in a radial channel blow $N_2$ uniformly in front of the orifice plate so that only sampled ions and this clean
$N_2$ gas pass into the vacuum chamber. Since the sampling flow rate of the mass spectrometer is ~0.5 LPM when using 0.3
mm orifice, the $N_2$ curtain flow is set to be 1 LPM to overflow the region surrounding the orifice. By applying voltages to
the ion source and the block, the ions can be efficiently guided into the mass spectrometer while neutral molecules such as
water vapor are prevented from entering by the $N_2$ curtain gas.

This TI inlet is suitable to all types of reagent ion chemistry, e.g. $NO_3^-$, $I^-$, and $H_3O^+$. Nitrate ion chemistry was used as the
reagent ion in these experiments, which is selective to highly oxidized molecules that have at least two hydroperoxy (-OOH)
groups or some other H-bond-donating groups (Hyttinen et al., 2015). HOM monomers, HOM dimers and highly oxidized
$RO_2$ radicals can also be measured using nitrate ion chemistry.




### 2.2.2 Other measurements

Ozone concentrations were measured with two ozone analyzers (model 106L, 2B Technology) at the inlet and outlet of the
flow tube. The sampling flow of each analyzer is 1 LPM. The two ozone analyzers were intercompared prior to the
experiments and the difference was within 5 ppbv when $[O_3] < 1000$ ppbv. A Scanning Mobility Particle Sizer (SMPS),
consisting of a Po210 bipolar neutralizer, a nano-Differential Mobility Analyzer (nano-DMA; model 3081, TSI, Inc.), and a
condensation particle counter (MCPC; model 1720, Brechtel Manufacturing) were used to measure the number-size
distribution of particles, which is later used to deduce the total particle number and mass concentrations (the latter assumes a
uniform density for organic particles of 1.2 g cm$^{-3}$). The sampling flow rate of the MCPC was 0.3 LPM and the sheath and
excess flows of the nano DMA were set to 3 LPM. The flow tube particle number-size distribution was measured without
further drying to get a more accurate measure of the actual particle surface area and volume, which are important for HOMs
partitioning, and also to prevent particle evaporation during the measurements.

### 2.3 Experimental conditions

Three monoterpenes were used in our experiments (see Table 1), α-pinene, limonene and Δ³-carene. Oxidation by ozone is
believed to dominate over other oxidation radicals (i.e., OH or $NO_3$) in forming secondary organic aerosol (SOA) under
atmospheric conditions (Atkinson and Arey, 2003). Ozonolysis of alkenes generates a substantial amount of OH, leading to
products that are produced by a combination of $O_3$ and OH oxidation. For some experiments, in order to isolate oxidation by
$O_3$, cyclohexane (see Table 1 for mixing ratios) was premixed with the monoterpene and added to the flow tube as an OH
scavenger. For other experiments, the combination of OH and $O_3$ chemistry were investigated to study atmospheric oxidation
chemistry more representative of ambient air. The "high concentration" experiments were conducted with similar mixing
ratios of monoterpene (~1100ppb) and $O_3$ (~900ppb). The "low concentration" experiments were conducted to study the
particle-free chemical processes with initial concentrations of monoterpenes and $O_3$ shown in Table 1. Since wall losses
should be comparable for different precursors as a function of RH, it was not taken into consideration in our analysis of
HOMs production.

### 2.4  HOMs volatility predictions

The SIMPOL.1 method (Pankow and Asher, 2008) and the molecular corridor method (Li et al., 2016) were used to predict
the saturation mass concentrations (C*) of some of the detected OH- and $O_3$-related HOMs. SIMPOL.1 is a group
contribution method and requires information on molecular structure, while the molecular corridor method only requires the



molecular formulae. Both methods are semi-empirical and based on volatility data from hundreds or thousands of
compounds. The calculated volatilities were then applied to the two-dimensional volatility basis set (2D-VBS) (Donahue et
al., 2012) to explore the likelihood that the products participate in the initial stages of nanoparticle growth.

**3    Results and discussion**
**3.1 TI-CIMS performance**
When comparing the TI inlet with the commercial nitrate inlet in measuring α-pinene ozonolysis products, both inlets
produced identical mass spectra. The sensitivities of both inlets to $H_2SO_4$ were determined using a home-built $H_2SO_4$
calibration system (Figure S2) based on the design of Kurten et al. (2012). Figure 3 summarizes the results of these
calibrations. The position of the ion source relative to the inlet orifice is critical for determining the sensitivity of the TI inlet.
When the ion source is placed 0.5 cm upstream along the sample flow axis and 5 cm away from the inlet orifice along the
reagent ion flow axis (configuration shown in Figure 2), the instrument is at its most sensitive. The calibration factors,
defined as $C = [H_2SO_4]/([HSO_4^-]/[NO_3^-])$ (Eisele and Tanner, 1993), for the TI in this position and the commercial inlet
were $3.25 \times 10^{10}$ molecules $cm^{-3}$ and $1.41 \times 10^{10}$ molecules $cm^{-3}$, respectively. The lower calibration factor for the TI is
attributed to the shorter reaction time, which we estimate to be ~80 ms. The total ion count (TIC) of the TI inlet is more than
5 times higher than the commercial inlet, so the overall sensitivity is better. The lower detection limit for sulfuric acid, which
is defined as three times the standard deviation of the background (Jokinen et al., 2012), is $9.3 \times 10^4$ molecules $cm^{-3}$ and
$1.26 \times 10^5$ molecules $cm^{-3}$ for the TI and commercial inlet, respectively.

After applying the $N_2$ curtain gas flow, the TIC recorded by the TI-CIMS decreased significantly. This was compensated for
by increasing the ion source and reaction chamber voltages that direct ions to the orifice (Figure S3). When RH>90%, the
reagent ion mass spectrum was dominated by water clusters $(H_2O)_m(HNO_3)_nNO_3^-$ (m=0~30, n=0~2) if no $N_2$ curtain flow
was applied. The reagent ions $NO_3^-$, $HNO_3NO_3^-$ and $(HNO_3)_2NO_3^-$ decreased as RH increased, with $[(HNO_3)_2NO_3^-]$ and
$[HNO_3NO_3^-]$ decreasing much faster than $[NO_3^-]$. In contrast, after 1 LPM $N_2$ curtain flow was applied to the inlet, most of
the water clusters were removed (Figure 4). The reagent ions, sample ions and TIC remained stable as RH increased, which
resulted in a reliable measurement of HOMs concentrations as a function of RH. The result that the $N_2$ curtain flow
eliminated water clustering to a large extent confirms that most of the water clusters in the spectrum were produced during
the free-jet expansion into vacuum instead of formed in the ion-molecular reagent region.



### 3.2 Identification of HOMs spectrum

Figure 5 shows the average mass spectra of the HOMs dimers and Figure S4 shows the average mass spectra of the HOMs monomer and $RO_2$ radicals for each of the six particle generation experiments. More than 400 peaks were identified in each spectrum, the majority of which were clusters with $NO_3^-$ or $HNO_3NO_3^-$. $[H_2SO_4]$ was ~$10^5$ molecules cm$^{-3}$ and was always less than 3% of the most abundant $C_{10}$ products, suggesting that sulfuric acid plays a negligible role in nucleation and cluster growth in our experiments. After subtracting the reagent ions ($NO_3^-$ or $HNO_3NO_3^-$), molecular formulae for organics with an odd number of H atoms were assigned to radicals, which are generally difficult to detect experimentally (Rissanen et al., 2015), and formulae with an even number of H atoms were assigned to closed-shell molecules. Most of the HOMs products from the three endocyclic monoterpenes were very similar, while the relative abundance of different HOMs was quite different, indicating similar reaction pathways but different branching ratios in the reaction mechanisms. The main products were $C_{5-10}H_{6-16}O_{3-10}$ for closed shell monomers and $RO_2$ and $C_{15-20}H_{22-34}O_{6-18}$ for closed shell dimers. Among these, $C_{10}$ and $C_{20}$ compounds were the most abundant. $C_{5-9}$ products could be formed from $O_3$ attack on the less reactive exocyclic carbon double bond or the decomposition of intermediate radicals. Some fragments were found to be unique for specific monoterpene precursors. For instance, $C_5H_6O_7$ (m/z 240) was much more abundant in α-pinene oxidation than from other two precursors, which might be a tri-carboxylic acid (Ehn et al., 2012).

Comparing total HOMs abundance for the three monoterpene oxidation reactions, limonene created the most, followed by α-pinene and then △³-carene. This is in qualitative agreement with prior studies (Jokinen et al., 2014; Ehn et al., 2014). The total dimer signal intensity was 15-30% of monomers for all three monoterpenes. Experiments with an OH scavenger generated fewer HOMs than those without OH scavengers.

As observed in previous studies, $C_{10}H_{15}O_{6,8,10,12}$ and $C_{10}H_{17}O_{5,7,9,11}$ comprised the $O_3$- and OH-related $RO_2$, respectively (Jokinen et al., 2014), while $C_{10}H_{14}O_{5,7,9,11}$ and $C_{10}H_{16}O_{6,8,10,12}$ comprised the $O_3$- and OH-related closed shell monomers, respectively (Ehn et al., 2014). When comparing the average spectra with and without OH scavenger, no obvious differences were seen for OH-related $RO_2$ or monomers (Figure S4). In contrast, for dimers we found that $C_{20}H_{32}O_{6-13}$ were more abundant in experiments without OH scavenger (Figure 5). The formation of these dimers can be explained by the reaction of one OH-related $RO_2$ with one $O_3$-derived $RO_2$ (see Section 3.5), and can therefore be considered as markers for combined OH and $O_3$ chemistry. As HOMs dimers are generally less volatile than monomers with identical O/C ratio, rapid production of dimers is believed to play a more important role in initial particle formation and growth (Zhang et al., 2015).



### 3.3 RH influence on HOMs generation

Figure 6 shows a time series of experimental parameters, particle size distribution, and key ions from the limonene ozonolysis experiment with OH scavenger (EXP. 2 in Table 1). The $O_3$ inlet and outlet concentrations were approximately constant with increasing RH (Figure 6a), indicating that RH did not significantly change $O_3$ levels in the flow tube. This also shows that the reactivity of the limonene with ozone does not change with RH. The number concentration of the generated particles decreased from $4.9 \times 10^6$ cm$^{-3}$ to $2.7 \times 10^6$ cm$^{-3}$ with increasing RH, while the peak of the number-size distribution increased slightly, due in part to water absorption. When RH was above 80%, both the integrated number and mass concentrations, which were calculated from the number-size distributions, decreased (Figure 6b).

Despite the change in particle number and mass concentrations with RH, the concentration of all the main HOMs, including $RO_2$, monomers and dimers, did not change for both OH- and $O_3$-derived products (Figure 6c). In fact, the only signals in the mass spectra that changed with RH corresponded to an increases associated with water clusters. The variations in HOMs concentrations can be explained by the competition between production and condensational losses. With the changes of particle concentration, the condensation sink should also change with RH. However, in this experiment, as in all other experiments, the surface area for existing particles ($1.8 \sim 4.1 \times 10^{-6}$ m$^2$) was much lower than the wall surface area of the flow tube ($\sim 0.38$ m$^2$). As almost all of the detected HOMs are ELVOCs or LVOCs (see Section 3.6), they are not likely to partition back to the gas phase after they encounter a wall. As a result, the main loss in the flow tube should be caused by wall loss, which does not change significantly with RH. To further test this hypothesis that wall losses dominated over condensation onto particles, particle free experiments were performed and, again, the detected HOMs signals did not change with RH (Figure 7).

### 3.4 RH influence on SOA generation

Figure 8 shows the integrated SOA particle number and mass concentrations. The generated SOA particle number and mass concentrations for limonene were ~5 times greater than for $\triangle^3$-carene and α-pinene. This is because the theoretical ozone reactivity of limonene is 3~5 times higher than the latter two and molar yield from limonene ozonolysis is also the highest. Peaks in the particle number-size distributions were between 40 and 70 nm (Figure S5). In most of the experiments, generated SOA mass concentrations increased or decreased slightly when RH increased from ~0% to ~60% and decreased as RH further increased to ~90%. The variability in particle mass concentration as a function of RH for different experiments can be attributed to combined effects of gas phase reactions, condensed phase reactions, as well as physical uptake of water, but all values were within a factor of two of each other. On the contrary, the particle number concentrations decreased by a





factor of 2~3 with increasing RH.

A number of studies have demonstrated different water and OH influences on the ozonolysis products of exocyclic and
endocyclic organic compounds. They have reported either suppressing (Bonn et al., 2002; Bonn and Moorgat, 2002) or
promoting (Jonsson et al., 2006; Jonsson et al., 2008) effects of water vapor on the particle formation processes during
ozonolysis of monoterpenes by measuring the number-size distributions of generated SOA particles with SMPS. The
discrepancies between different results could be attributed to the different experiment setups, e.g., monoterpene and $O_3$
concentration, temperature, RH range, OH scavengers, reaction time, and so on. Specifically, our results are in good
agreement with those of Bonn et al., who studied SOA generation from the ozonolysis of endocyclic monoterpenes (Bonn et
al., 2002). In that study, SOA number concentrations decreased by a factor of 1.1~2.5 as RH increased, while the variation in
volume concentrations was negligible (within ±10%). They concluded that water's influence on non-volatile products, which
are responsible for the initial steps of nucleation, was much larger than its influence for semi-volatile compounds which
mainly determined the final volume concentrations of particles. Thus, it was highly suspected that water influenced new
particle formation through influencing the generation of NPF precursors. However, our measurements indicate that at least
the formation of the detected HOMs is independent of water vapor concentrations. There may be other species that are
crucial to the initial steps of NPF and are affected by water vapor but are not detected by nitrate CIMS (see section 3.5).
Another possible explanation is that a fraction of HOMs cluster with water at high RH in such a way that they may no longer
be able to participate in further cluster formation, thereby suppressing NPF. If the CIMS measurement only detected the
declustered molecule, then such a mechanism may still be consistent with our observations.

**3.5 Possible water-relevant $C_{10}$ and $C_{20}$ HOMs formation pathways**
Although the oxidation of BVOCs has been widely studied, it has mostly been constrained to the early stages (first and
second generation intermediates) and many uncertainties still exist (Johnson and Marston, 2008; Isaacman-VanWertz et al.,
2018; Atkinson and Arey, 2003). The first step of ozonolysis for the three BVOCs (α-pinene, △³-carene and limonene) is
ozone attack on the endocyclic carbon double bond to form a primary ozonide. Figure 9 shows the $O_3$-initiated oxidation
pathways of α-pinene that may be related to the detected $C_{10}$ and $C_{20}$ HOMs for representative isomers. The primary ozonide
rapidly transforms to two excited Criegee intermediates (eCIs), one of which (branching ratio= 0.4) (Kamens et al., 1999) is
shown in Figure 9. The reaction pathways of the eCI are complex, the most important two under ambient and most chamber
conditions are the sCI channel (reaction I) and the hydroperoxide channel (reaction II) (Bonn et al., 2002). The sCI either
reacts with aldehydes to form a secondary ozonide (when the aldehyde is $C_{10}$, then the formed SOZ is $C_{20}$ and is marked as





sCI-$C_{10}$) or with water or other acidic compounds such as alcohols and carboxylic acids to form hydroxy-hydroperoxide,
which then decomposes to carboxylic acids or aldehydes. For α-pinene, the main decomposition product is pinonic acid. In
the hydroperoxide channel (reaction II), the formed hydroperoxide quickly decomposes to a first generation alkyl radical (R)
and OH (Johnson and Marston, 2008). R reacts with $O_2$ immediately to form the first generation $RO_2$, which can undergo
numerous reactions, including reaction with $HO_2$, $R'O_2$ and autoxidation. The reaction with $HO_2$ mainly forms
hydroperoxides, with a small fraction forming peroxides or carbonyl-containing compounds. However, the carbonyl cannot
be formed in most cases since the C atom bonded to O-O does not have available electrons for the carbonyl π-bond. When
reacted with another $R'O_2$, either ROOR' or an alkoxy radical (RO) or a carbonyl and a hydroperoxide are formed. The RO
can undergo isomerization, or form a carbonyl and $HO_2$, for which the branching ratios are extremely difficult to evaluate.
RO can also undergo decomposition, which is one of the pathways to form $C_5\sim C_9$. The autoxidation process is key to HOMs
formation. Each autoxidation step adds two O atoms to the molecule and thus increases the oxidation state very rapidly. The
competition between autoxidation processes and bimolecular reactions ($RO_2$ reactions with $R'O_2$ or $HO_2$) determines the
ultimate oxidation state of the products (Barsanti et al., 2017; Crounse et al., 2013; Rissanen et al., 2015).

OH can be generated in the ozonolysis of alkenes and the yield is near unity (Atkinson, 1997). The reaction of OH with
α-pinene directly forms first generation R and then $RO_2$; one possible structure for this $RO_2$ (branching ratio = 0.44), formed
from OH addition to the double bond (Berndt et al., 2016), is shown in Figure 9. However, the formed $RO_2$ ($C_{10}H_{17}O_{2m+1}$) are
not the same $RO_2$ as those formed through ozonolysis ($C_{10}H_{15}O_{2n+2}$) (McVay et al., 2016). Accordingly, the structure and
composition of $C_{10}$ and $C_{20}$ HOMs formed from OH or $O_3$ chemistry are different, and so too are their potential impacts on
NPF. The combined OH- and $O_3$-derived dimers ($C_{10}H_{32}O_{2(m+n)+3}$), formed by collision of an OH-derived $RO_2$ with an
$O_3$-derived $RO_2$, were only observed in ozonolysis experiments without OH scavenger.

The $RO_2$ autoxidation pathway explains most of the observed $C_{10}$ and $C_{20}$ compounds in the mass spectra. One exception to
this is that $C_{10}H_{18}O_{2m+1}$, $C_{10}H_{18}O_{2m}$ and $C_{20}H_{34}O_{2(m+m')}$ were not observed in the spectrum whereas in experiments performed
by Berndt et al., in which OH oxidation for α-pinene was studied, $C_{10}H_{18}O_{2m}$ and $C_{20}H_{34}O_{2(m+m')}$ dominated the mass
spectrum (Berndt et al., 2016). This could be explained by a low OH/$O_3$ ratio in our experiments, since unlike Berndt et al.
we did not provide a source of OH to the flow tube.

Despite this close agreement achieved by the $RO_2$ autoxidation mechanism and the observed mass spectra in our study, prior
studies suggest that other potential pathways cannot be excluded. An accretion product involving sCI is one possibility





(Barsanti et al., 2017). It is possible that sCI reacts with long-chain carboxylic acids or carbonyls, such as those with 10
carbon atoms, forming in this instance anhydrides (sCI-$C_{10}$, reaction IV) or secondary ozonides (sCI-$C_{10}$, reaction V) with
vapor pressures lower than $10^{-15}$ torr (Kamens et al., 1999; Tobias and Ziemann, 2001; Bonn et al., 2002). The formation of
anhydride is more likely in condensed phase, whereas there is also a possibility it can also happen in gas phase (Kamens et
al., 1999). However, it is unknown whether these sCI-$C_{10}$ can be detected using nitrate-CIMS as they may lack hydrogen
bond donor moieties. The semi-volatile pinonic acid can also form HOMs after further oxidation by OH (Ehn et al., 2014),
provided that excess α-pinene is not present to compete with pinonic acid for the generated OH.

Water vapor's influence on HOMs formation can be direct or indirect. For monoterpene oxidation, the direct participation of
water vapor is to react with sCI, favoring the formation of the hydroperoxide and its decomposition products (reaction III)
over the secondary ozonides (sCI-$C_{10}$, reaction IV) or possible anhydrides (sCI-$C_{10}$, reaction V). Since the formation of
sCI-$C_{10}$ is more likely to contribute to NPF than the products from sCI and water vapor (Kamens et al., 1999; Tobias et al.,
2000), a decrease in low volatility sCI-$C_{10}$ with high RH could explain the decreasing SOA number concentrations in our
experiment. It has been shown previously that OH yields from the reactions of $O_3$ with a series of monoterpenes were not
affected by the presence of water vapor (Atkinson et al., 1992; Aschmann et al., 2002), which implies that the hydroperoxide
channel (reaction II) are similarly unaffected by water. Since the detected HOMs in our experiments were RH-independent,
we conclude that all the detected HOMs were formed from hydroperoxide channel (reaction II) and not via the sCI channel
(reaction I). Similarly, the detected HOMs were not likely to form through the hydration reaction (Equation 1) (Ehn et al.,
2012), which is supposed to increase with increasing RH.
$$R - CHO + H_2O \rightarrow R - CH(OH)_2 \quad .$$   (Equation 1)

The indirect water effect on HOMs formation includes the water influence on $HO_2$ fate. As water promotes $HO_2$ self-reaction
(Equation 2), reaction of $HO_2$ with $RO_2$ should decrease and the related HOM monomers should likewise decrease with
increasing RH. However, as the formation of both HOM monomers and dimers was not affected by $H_2O$, it was likely that
water does not significantly increase $HO_2$ self-reaction or that $HO_2$ chemistry was not important in our experiments.
$$2HO_2 \xrightarrow{M, H_2O} H_2O_2 + O_2$$   (Equation 2)

**3.6 Volatility predictions**
The volatility of the gas phase products is one of the most important properties that determines whether a compound
contributes to the formation, initial growth or further growth of SOA particles (Donahue et al., 2012; Kroll et al., 2011). As





the products with identical elemental composition can be formed from different bimolecular reactions of the intermediate
RO$_2$, it is difficult to predict their exact structures. For the current study, the number of different structural and functional
groups (e.g., aromatic rings, carbon double bonds, aldehydes, carbonyls, hydroxyls, ethers, hydroperoxyls) was estimated
and used to derive saturation vapor pressure using SIMPOL.1 (Table S1). To simplify the calculation, the functional groups
used here were directly predicted from the proposed formation pathways in Figure 10 and did not include intramolecular
isomerization, although that may be important in some situations. For example, one of the ROOH can be replaced with an
endo-peroxide via ring closure of unsaturated RO$_2$ (Berndt et al., 2016). To figure out the possible bias introduced by this
simplification, the result was compared to those obtained using the Molecular Corridor method (Li et al., 2016), the latter of
which does not require information on functional groups.

Figure 10 shows the predicted saturation mass concentrations, C*, of the main C$_{10}$ and C$_{20}$ closed shell products. The
difference of C* predicted from the two methods was within one order of magnitude for C$_{20}$ HOMs, and 3~4 orders of
magnitude for C$_{10}$ HOMs. The difference was due to the abundance of –OOH and –OH moieties, which contribute more to
lowering saturation vapor pressures than other functional groups (Table S1). Despite these differences, nearly all of the C$_{20}$
HOMs can be classified as ELVOCs, while C$_{10}$ products were mostly LVOCs. Typically, the volatilities of O$_3$-derived C$_{20}$
HOMs were less than OH-related HOMs, whereas for those with identical oxidation states of carbon, $\overline{OSc}$ (defined as
2O/C-H/C) (Kroll et al., 2011), such as C$_{20}$H$_{30}$O$_{10}$ (O$_3$-derived dimer), C$_{20}$H$_{32}$O$_{11}$ (OH and O$_3$ combined dimer), C$_{20}$H$_{34}$O$_{12}$
(OH-derived dimer), OH-derived C$_{20}$ HOMs have lower volatilities than O$_3$-derived HOMs due to a greater number of
oxygen atoms in the former.
**4 Conclusions**
The RH influence on HOMs formation and NPF during monoterpene oxidation was explored in this study. HOMs were
detected with a TI-CIMS, using nitrate as reagent ions; C$_{10}$ and C$_{20}$ dominated the spectra. There are mainly three potential
paths for water vapor influence on the formation of C$_{10}$ and C$_{20}$ HOMs. One is water reacting with sCI (Equation 1), thereby
influencing the branching ratio between formation of more volatile compounds decomposed from hydroxyl hydroperoxide,
such as pinonic acid, and accretion products with sCI such as secondary ozonide (sCI-C$_{10}$) and anhydride (sCI-C$_{10}$). The
second hypothesized water influence is on the HOMs formed from hydration reactions (Equation 2). The third is that water
increases the rate of self-reaction of HO$_2$ (Equation 3), thus indirectly impacts the loss pathways of RO$_2$. Our experimental
results, both with high particle loading and particle-free conditions, demonstrated that neither the detected HOMs species nor
their signal abundance changed significantly with RH. This indicates that the detected HOMs, which can mostly be





explained by $RO_2$ autoxidation, must be formed from water-independent pathways rather than by those reactions mentioned
above. One implication of this result is that $HO_2$ self-reaction was not significantly promoted by water or that the $RO_2$
reaction with $HO_2$ was not be significant in our system, but instead that $RO_2$ reacts with another peroxy radical, $R'O_2$, to
generate both closed shell monomers and dimers. Another implication is that the sCI pathway is not responsible for the
generation of the detected HOMs while the role of sCI-related HOMs (SOZ or anhydride) formation by accretion with long
chain products, which may not be detected with nitrate CIMS, may be important in causing the decrease in SOA number
concentrations with increased RH. Another possible explanation for the decreasing SOA number concentration is that water
may cluster with HOMs and suppress NPF.

The detected HOMs, which could mostly be explained by autoxidation of $RO_2$ followed by reactions with $R'O_2$ or $HO_2$, were
distinguished as OH-related, $O_3$-related $RO_2$, closed shell HOM monomers, and HOM dimers. The volatility of the identified
products were estimated with the SIMPOL.1 group contribution method and with the molecular corridor technique. That
analysis confirmed that $C_{20}$ closed shell products have significantly lower volatility compared to $C_{10}$ products and are thus
more likely to contribute to NPF. Pure $O_3$ chemistry produced lower volatility $C_{20}$ closed shell products compared to
processes that were influenced or dominated by OH. As a result, $O_3$ chemistry is suspected to be more likely to lead to NPF
than OH chemistry, given the same level of oxidants and VOCs precursors.

**Acknowledgements**
This research was supported by the US Department of Energy's Atmospheric System Research program under grant no.
DESC0014469, the US National Science Foundation under grant no. AGS-1762098, and by the National Key R&D Program
of China under grant no. 2017YFC0209503. XL thanks the financial support from the State Scholarship Fund managed by
Chinese Scholarship Council (CSC). We thank Hayley Glicker, Deanna Caroline Myers, Michael Lawler, and Danielle
Draper for their kind help.

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



**Table 1. Experiment conditions and products.**

| Precursor | Exp (#) | Monoterpene (ppbv) | $O_3$ (ppbv) | Cyclo-hexane (ppmv) | Initial rate[1] ($10^8$ molecules cm$^{-3}$ s$^{-1}$) | $O_3$ consumption[2] (ppb) | SOA[3] ($\mu$g m$^{-3}$) |
|---|---|---|---|---|---|---|---|
| Limonene | 1 | 1085 | 900±10 | 0 | 1410 | 159-166 | 138-208 |
| | 2 | 1085 | 900±10 | 217 | 1410 | 139-150 | 81-147 |
| | 3 | 54 | 350±5 | 0 | 27.3 | 34-41 | 0 |
| α-pinene | 4 | 1111 | 900±10 | 0 | 625 | 103-110 | 761-1042 |
| | 5 | 1111 | 900±10 | 222 | 625 | 93-102 | 414-735 |
| | 6 | 54 | 350±5 | 0 | 11.8 | 23-30 | 0 |
| $\triangle^3$-carene | 7 | 1111 | 900±10 | 0 | 267 | 72-89 | 55-93 |
| | 8 | 1111 | 900±10 | 222 | 267 | 70-86 | 34-92 |
| | 9 | 54 | 350±5 | 0 | 5.05 | 11-16 | 0 |

[1]**At room temperature (298K), the rate coefficients for limonene, α-pinene and $\triangle^3$-carene to react with $O_3$ were $200\times10^{-18}$,**
**$86.6\times10^{-18}$, $37\times10^{-18}$ cm$^3$ molecule$^{-1}$ s$^{-1}$, respectively.**
[2]**$O_3$ consumption values were calculated from the difference between inlet and outlet $O_3$ concentrations.**
[3]**SOA mass concentrations were calculated from SMPS-measured volume concentrations and an assumed organic effective density**
**(1.2 g cm$^{-3}$).**





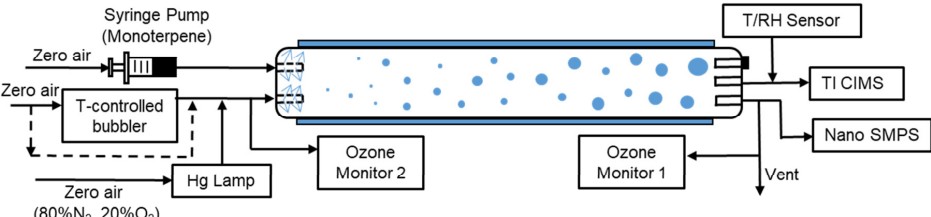


**Figure 1. Experiment setup for the flow tube experiments. The 8.5 L flow tube was placed at a temperature-controlled room**
**(21±1°C) and covered. The total flowrate was 8.5 LPM. The RH was adjusted by mixing temperature controlled bubbler flow with**
**dry zero air.**

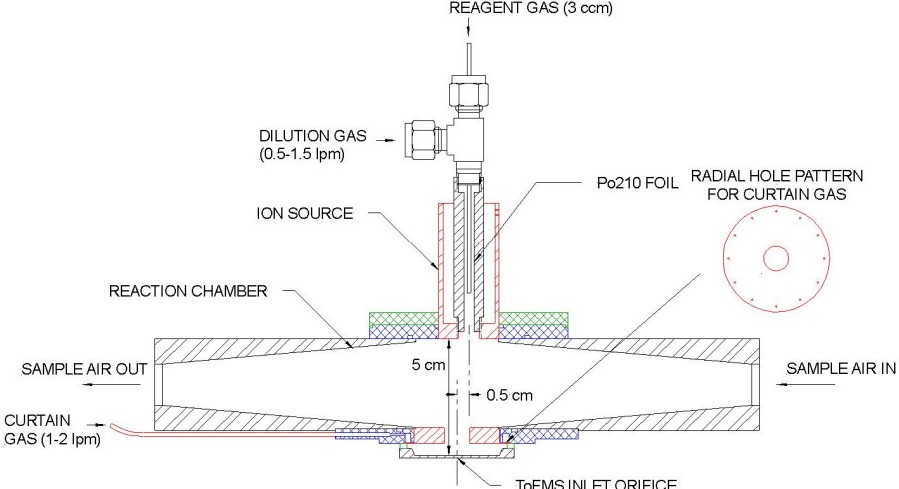


**Figure 2. Schematic of the transverse ionization (TI) inlet, showing the $N_2$ curtain gas configuration. The relative position of the**
**ion source to the inlet orifice is adjustable. The configuration shown here is the most sensitive in calibrations with $H_2SO_4$ (see**
**Section 3.1).**



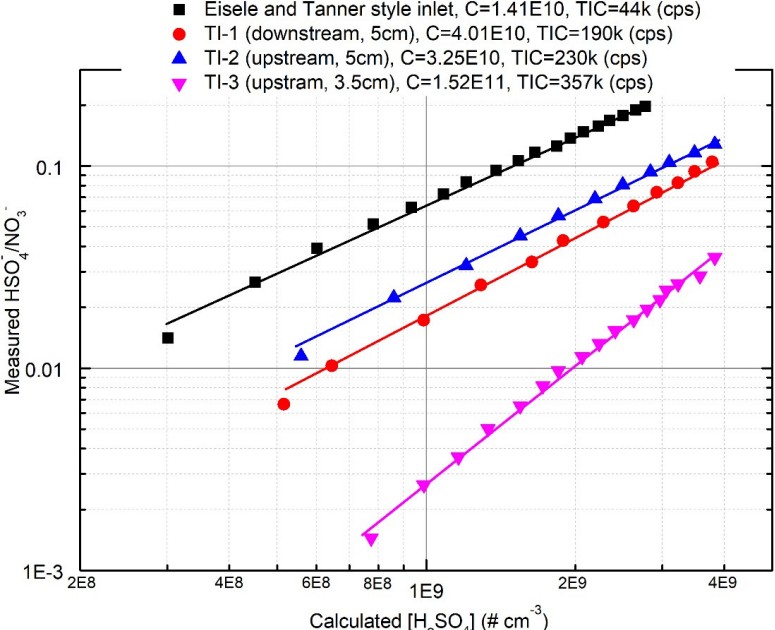


**Figure 3. Comparison of the sensitivities for the two inlets to $H_2SO_4$. The calibration process followed that reported by Kurten et al.**


**(2012) and is discussed in detail in the supplementary material. TI-1, 2, 3 represent different locations of the ion source relative to**


**the inlet orifice of the mass spectrometer. "upstream" and "downstream" indicated 0.5 cm upstream or downstream along the**


**sample flow axis and "3.5 cm" and "5 cm" indicate 3.5 or 5 cm away from the inlet orifice along the reagent ion flow axis.**


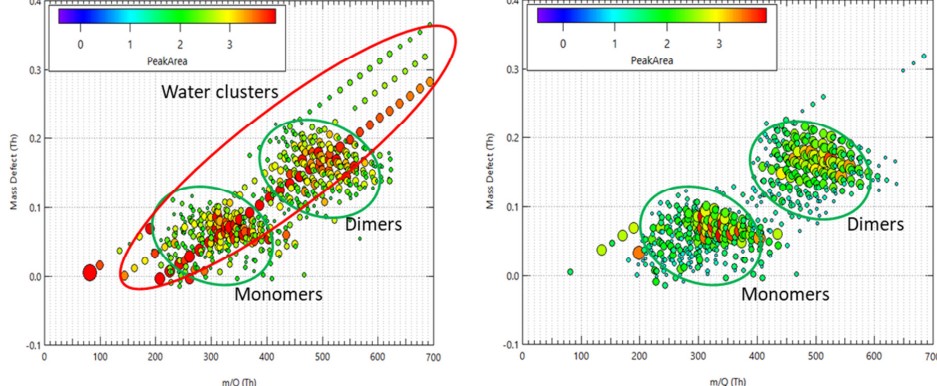


**Figure 4. Mass defect plots of α-pinene ozonolysis HOMs with 0 LPM (left) and 1 LPM (right) $N_2$ curtain gas flow when RH> 85%,**


**with monomer and dimer HOMs circled in green. The most intense ions comprising 60% of the total ion count are plotted for**


**clarity. $H_2O$ clusters $(H_2O)_m(HNO_3)_nNO_3^-$ (m=1~30, n=0~2) are circled in red in the left plot and are notably absent with the**


**application of the $N_2$ curtain gas. $(HNO_3)_2NO_3^-$ and $HNO_3NO_3^-$ are much more likely to cluster with $H_2O$ than $NO_3^-$.**





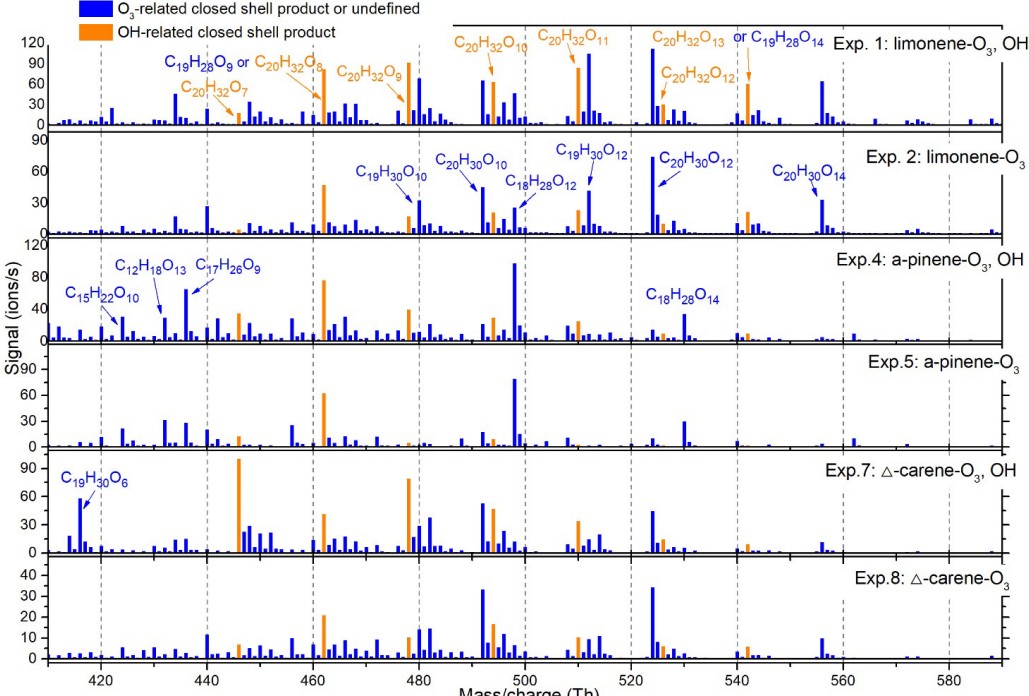

**Figure 5. Average dimer mass spectrum in each of the particle generation experiments. The OH- and O₃-derived species were**

**distinguished by comparing relative abundance of experiments with and without OH scavenger. All the peaks shown were in the**

**form of adducts with NO₃⁻ or HNO₃NO₃⁻ reagent ions.**

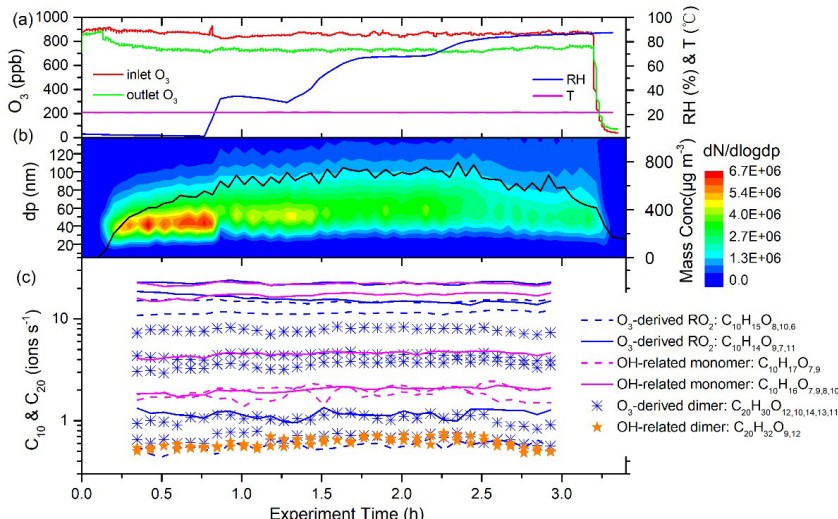

**Figure 6. Time series of experimental parameters, particle size distribution, and key ions during EXP. 2 (limonene oxidized by O₃**

**without OH scavenger). (a) inlet and outlet O₃ concentrations, temperature, and RH; (b) Particle size distribution and integrated**



**mass concentrations (assuming effective density is 1.2 g cm⁻³); (c) Some of the main HOMs detected by TI-CIMS with NO₃⁻ reagent**
**ion. The subscript oxygen numbers in the formulae were ranked (left-to-right) according to signal abundance of the corresponding**
**molecule.**

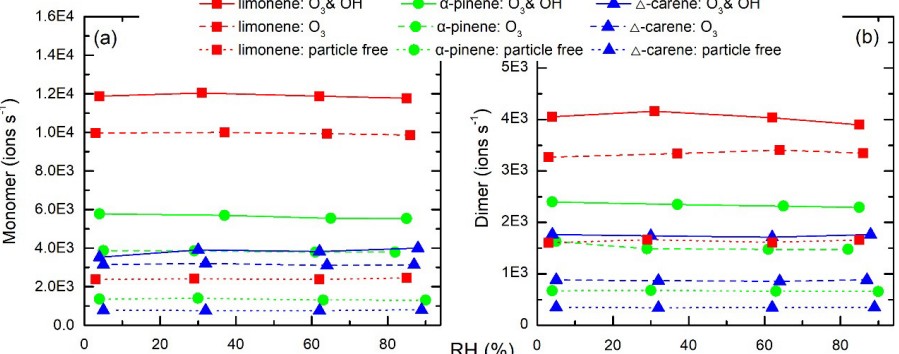


**Figure 7. Average (a) monomer and (b) dimer HOMs signal intensity (ions s⁻¹) as a function of RH in each experiment. Monomer**
**signals were the sum of C₅₋₁₀ molecules and dimer signals were the sum of C₁₅₋₂₀ molecules. No obvious signal change was seen for**
**increasing RH in any of the experiments.**

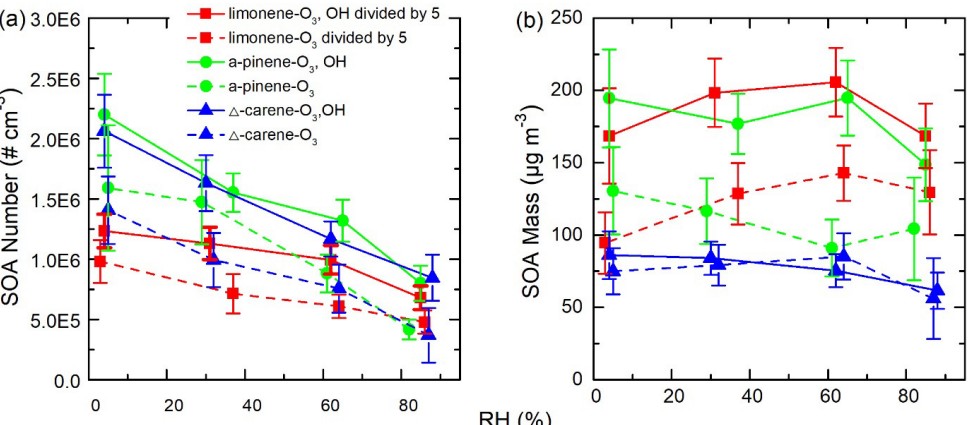


**Figure 8. SOA (a) number and (b) mass concentrations as a function of RH during different experiments. The error bars were**
**calculated using both the statistical errors of all individual size distributions during each RH stage and assuming a systematic CPC**
**counting error of 10%).**






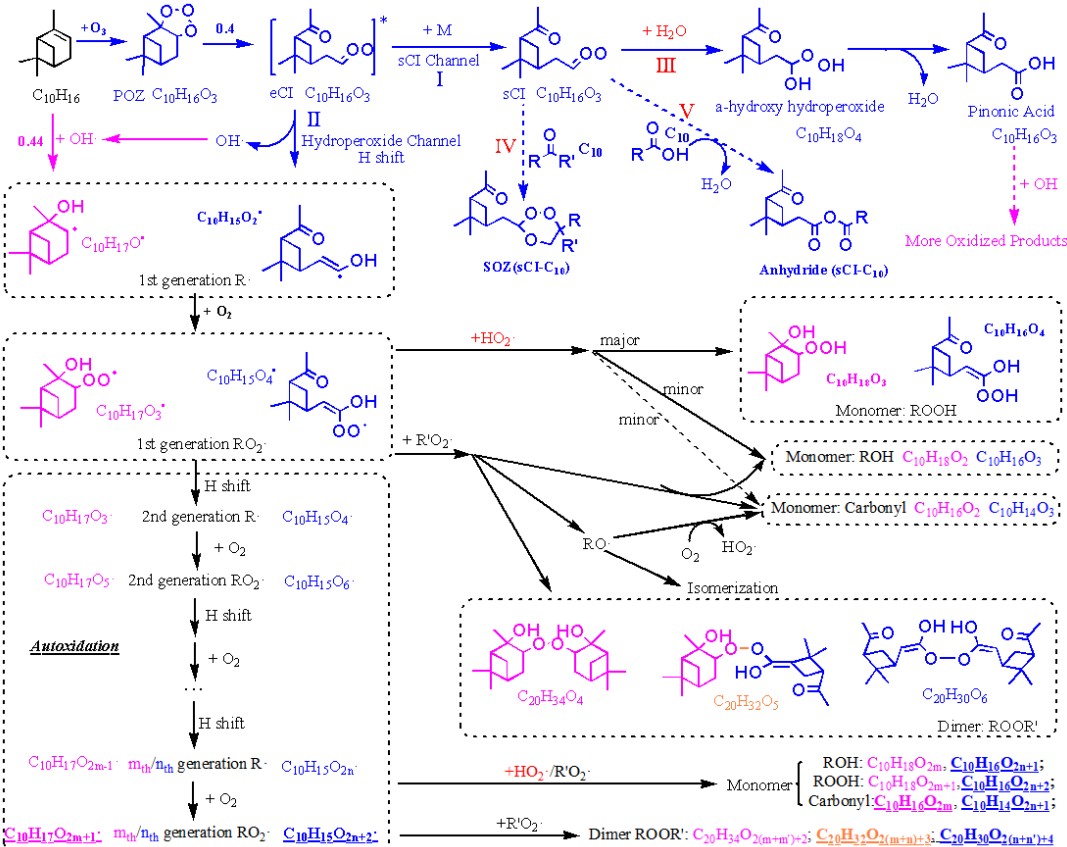

**Figure 9. Proposed key steps in the formation of the representative $C_{10}$ and $C_{20}$ closed shell products from α-pinene oxidation and**
**possible water vapor influence. Dashed lines represent pathways that may or may not happen, depending on the situation. Pink**
**and blue colors represented the pathways or products from $O_3$ and OH oxidation, respectively. Common pathways or products are**
**indicated in black type. Orange colors represented the combined products of $O_3$ and OH chemistry. Red colors highlight the direct**
**or indirect influence of water. Underlined formulae were the main products observed from the mass spectrum. $C_{10}H_{18}O_{2m}$ and**
**$C_{20}H_{34}O_{2(m+m')}$ were not observed in our spectrum, but they dominated the spectrum in other reported experiments where extra**
**OH was generated (Berndt et al., 2016).**





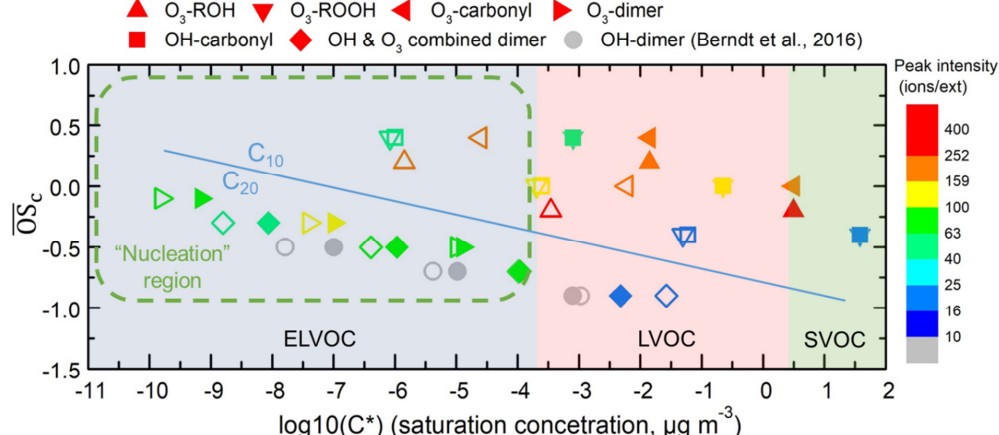


**Figure 10. Vapor saturation mass concentration C\* (T=298 K) of the major $C_{10}$ and $C_{20}$ closed shell products were predicted with**
**SIMPOL.1 (open points) (Pankow and Asher, 2008) and Molecular Corridor method (filled points) (Li et al., 2016). $O_3$-derived,**
**OH-related and OH-derived monomers and dimers are presented in different shapes. The peak intensity, represented by color, is**
**from Exp.1 (limonene oxidation without OH scavenger). The gray points, which represent OH-derived dimers, are dominating**
**products in OH initiated oxidation experiments (Berndt et al., 2016) while not observed in this study. Data used in this figure are**
**given in Table S1. The nucleation region is from Donahue et al. (2013).**