# Peer review of "Relative Humidity Effect on the Formation of Highly Oxidized Molecules and"

_Atmospheric Chemistry and Physics, 2018_

## Referee Comment (RC1) · Anonymous Referee #1 · 22 Oct 2018

Ms.No.: acp-2018-898

The authors describe experimental findings from the ozonolysis of a-pinene, limonene and 3-carene from a flow through experiment conducted in the RH range of 3 – 90%. Runs have been carried out with the intention to study the RH-dependence of HOM formation of the terpenes and the resulting nucleation and particle growth. While HOM formation was found to be independent of RH, particle formation was clearly pushed back for rising RH. Some speculative explanations for that are presented.
From my perspective, very interesting is the new transverse ionization inlet with the curtain gas unit, which could be an alternative to the commonly used Boulder-type inlet, as well as the experimental fact that HOM formation is totally free of water effects. I think that the manuscript meets the criteria for ACP and should be published in this journal. Some minor points should be considered before final acceptance is recommended:

1) Line 78-80: The authors mean that the intra-molecular H-shift, or RO2 isomerization, is characterized by a noticeable barrier making this unimolecular step clearly faster with rising temperature. I think it´s not good to say "autoxidation" has a barrier. Autoxidation stands more for the whole process.

2) Line 85-86: The Boulder-type nitrate-CIMS by Eisele and Tanner does not suffer from a general problem with water vapour. Only in the case of relatively high RH in the reaction gas water cluster formation during gas expansion can disturb the analysis. Otherwise it works fine. That should be clearly stated at this point. Or have the authors other observations?

3) Line 90: In abstract a RH range of 3 – 90% is stated, and here 0 – 90%.

4) Line 114: It would be fine to have a table that compares parameters of the TI inlet with those of the commonly used boulder-type inlet, e.g. reaction times of the IMR, flows, HNO3 concentrations, TIC, detection limits, an estimate of wall losses for RO2s and closed-shell products, etc. Spectra of the same reaction gas recorded with both inlets would provide an impression how good the TI inlet works. Maybe the authors should think about a separate paper describing the TI inlet in detail. Could be important for the community.

5) Line 223: Can the authors derive HOM yields as a result of their experiments?

6) Line 228-229: Reaction of OH-derived RO2 radicals, C10H17Ox, with HO2 leads to H18 products, ROOH.

7) Line 258: Figure S5 should be given in the main body along with Fig.8. From my perspective, total SOA mass is almost water-independent within the experimental errors in the whole RH range while particle number drops down by a factor of about two.

8) Line 319: Here, reagent ion dependent sensitivity for different product classes should be mentioned that could lead to different results for different reagent ions.

---

## Referee Comment (RC2) · Anonymous Referee #2 · 26 Oct 2018

This study examines the influence of relative humidity (RH) on the formation of HOM and aerosol from the ozonolysis of three different monoterpenes. HOM have been shown to be central for particle formation over the last years, and therefore knowing their yields under different atmospherically relevant conditions is important. The influence of RH has clearly not been addressed in earlier studies, making this work timely and appropriate.

The paper is well-written and the presentation of results is clear and straight-forward. The strength of the paper lies in the finding that HOM yields do not change as a function of RH, and this analysis by itself makes the manuscript worthy of publishing in ACP. The

results and subsequent discussion on number and mass formation seem much more speculative. Overall, I have several comments that the authors need to address before potential publication.

Major comments:

1. RH control. In section 2.1, the authors describe that the RH in the flowtube was achieved by mixing a humidified and a dry flow. The humidified flow was 6.5 LPM while the total in the flow tube was 8.5 LPM. Assuming these flows were at the same temperature (which one would hope) when entering the flow tube, then even if the humidified flow was fully saturated at RH=100%, then the maximum achievable RH should be 6.5/8.5 = 76%. However, the authors state that RH was probed up to 92%. I do not see how this is possible, unless the humidified flow indeed was several degrees warmer when entering the flow tube. If this were the case, it would not be surprising that particle number formation decreased, as nucleation is an extremely temperature sensitive process.

2. In Figure 6b, when RH is 0, the mass concentration is steadily increasing over the course of nearly one hour. This is very surprising considering that the residence time in the flow tube is 1 minute. Clearly there is some memory effect, and this is not currently discussed at all in the manuscript. Accumulation of semi-volatile material on walls is a common phenomenon, and can perhaps explain this behavior. Whatever the reason, it puts very high uncertainties on the results, considering that RH may influence the rate of (re-)evaporation from the walls. Are the measured SOA mass changes really significant when considering this?

3. Particle number concentration was in excess of 1e6 cmˆ-3. At such high concentrations, couldn't nucleation "shut down" at a certain point just because the condensation sink starts to overwhelm the collisions between nucleating agents? Alternatively, the ultimately measured particle number might mainly be limited by coagulation between newly nucleated clusters/particles. While I am not an expert on nucleation, the extreme particle number concentrations raise many questions about how far the absolute values can be compared directly, as done by the authors.

4. Related to the previous point, the authors state that since the aerosol surface area is so much lower than the flow tube wall area, the dominant loss term for HOM will be wall loss. This statement seems completely illogical. How are the particles even formed in the flow tube if all ELVOC or LVOC would condense on the walls? The authors need to calculate the actual condensation sink (CS) produced by the particles and I think they will find that the lifetime of HOM due to CS is on the order of seconds, while wall loss will be a slower process. Calculating a coagulation sink in addition might also help address my earlier comment nr 3.

5. Section 3.6 discusses volatility estimates of HOM. However, it seems the authors are not aware of the work by Kurtén et al. (2016), entitled "$\alpha$-Pinene Autoxidation Products May Not Have Extremely Low Saturation Vapor Pressures Despite High O:C Ratios". This work certainly needs to be discussed in conjunction with this section, as it raises considerable doubts about the applicability of SIMPOL to vapor pressure estimates of HOM. Using SIMPOL in this work is still completely appropriate, but the large uncertainties should be noted. Also other parts of section 3.6 are questionable, but primarily the authors need to revisit the discussion about oxidation state OS_C. The formula they utilize (2*O:C-H:C) is not applicable in the case that molecules contain (hydro)peroxide functionalities. And according to Table S1, the authors assume this is the case for every molecule under discussion. This will require rewriting all parts where OS_C is discussed.

Minor comments

6. Section 2.1 describes the flow tube setup, but the description is in many places unclear or ambiguous. For the flows, 0.5 LPM is diluted by 6.5 LPM and then mixed with 2.5 LPM. This adds up to 9.5 LPM, but the total flow was apparently 8.5 LPM? There also seems to be different types of zero air used, although they are named the

same. Is some of it bottled synthetic air? Finally, the authors should clarify "Gas inlets to the flow tube were made from Teflon tubing that were capped and drilled with small holes" better, since as written, it remains unclear to me exactly how this part of the setup looked.

7. There is variability in earlier literature about how to write the plural of HOM. In some cases also the plural is "HOM", while in other studies they are "HOMs". However, the usage in many places in this work seems questionable. "HOMs dimers", "HOMs volatility", "HOMs production", etc, sound incorrect to me, and HOM should not be in plural form in these cases.

8. In the conclusions, lines 378-383 present various ways through which water could affect HOM formation, but then this is followed by a sentence saying that none of them actually take place. Are those lines then really needed, or are they just more likely to confuse a reader?

9. Line 139, the inlet flow to the mass spectrometer is given as 0.5 LPM, but is it not closer to 0.7 LPM? This is given in some earlier studies, and is closer to the theoretical value for a 0.3 mm orifice.

10. Line 304. This is not a very clear sentence, but if I understand it correctly, the hydroperoxide should be an alcohol?

11. Figure 3. The legend says "upstram".

12. Figure 4 needs larger axis labels.

13. Figure 6c. The legend for the lines is incorrect. O3 radicals are given twice, as are OH monomers. They should be mixed.

14. Figure 9. The molecule in blue in the top-most box: Is the keto form not more commonly depicted than the enol form given here?

15. There are references to "Kurten et al (2012)" and "Kuerten et al (2016)". As these,

to my understanding, are the same person, they should be spelled the same.

16. While the likelihood of misunderstandings about what particles are studied here is low, the word "aerosol" is not used before in section 2.3. I suggest to include it at least once at a much earlier stage.
* * *

---

## Referee Comment (RC3) · Anonymous Referee #3 · 4 Nov 2018

This manuscript describes the RH effects on HOMs formed from oxidation reactions (ozone and OH) of monoterpernes and on the SOA formation. The authors used HrTOF-CIMS to measure gas phase HOMs (monomers and dimers). The main conclusion is that because HOMs are not affected by RH under the present experimental conditions, formation pathways of HOMs may not include water. The review believes this is a reasonable explanation, considering that the autoxidation reactions take place in the condensed phase and at high temperatures, where water is less available. Considering the increasingly important roles of HOMs in NPF and SOA formation, this conclusion is useful to the community.

[Figure]

Since the measured particles sizes in this study are between 20 – 100 nm, it would be difficult to discuss the contribution of RH on NPF from these sizes. It is more likely that the precursors of nucleation (or NPF) and 20 - 100 nm particles have different volatilities and they may be even different precursors. Therefore, the RH effects on these aerosol particles are different from what the community considers regarding to RH effects on NPF (stated in Introduction). Rather, the results shown here indicate the effects on SOA formation. So, I would suggest reorganize Introduction and results/discussion to focus on HOMs. Since RH varies in a large range in the atmosphere, it is still important to understand the effects of RH on HOMs formation.

(1) Lines 38-42: The Amazon NPF or lack of it has little to do with RH, rather it is due to high condensation sink or the lack of nucleation precursors. (2) Line 60: Include Yu, H., et al. (2012). "Effects of amines on formation of sub-3 nm particles and their subsequent growth." Geophys. Res. Lett. 39: Doi: 10.1029/2011gl050099. (3) Section 2.4. Indicate the exact equation that is used to calculate saturation mass concentrations (C*). Later, it is stated that the functional groups are also considered, so it would be useful to show the calculation procedure. (4) Line 210: What is the source of e5 /cc of sulfuric acid? (5) Line 231: Dimers are more abundant from OH oxidation. This is an interesting result, considering that HOMs formed from ozonolysis are found mostly during the nighttime, where NPF does not take place. So this may explain the importance of dimers on aerosol nucleation. (6) Lines 251-252: Show condensation sink, instead of surface area. The size distribution shows aerosol sizes are 20-100 nm, so condensation actually takes place effectively. (7) Lines 260-261: Indicate yields. (8) Line 306: Decomposition to C5 HOMs. Is this new?

---

## Author Comment (AC1) · 18 Dec 2018

**Responses to Reviewers' Comments on Manuscript ACPD-2018-898**

**(Relative Humidity Effect on the Formation of Highly Oxidized Molecules and New Particles during Monoterpene Oxidation)**

We are grateful for the reviewers' comments and we feel that our responses to these will greatly improve this manuscript. We have addressed the comments in the following paragraphs and made corresponding changes in the revised manuscript. Reviewer comments are shown as *blue italic text* followed by our responses. Changes are shown as underlined text in our responses and highlighted in the revised manuscript, the latter of which (along with supplemental information) is attached to the end of this document.

**Reviewer #1:**

*The authors describe experimental findings from the ozonolysis of a-pinene, limonene and 3-carene from a flow through experiment conducted in the RH range of 3-90%. Runs have been carried out with the intention to study the RH-dependence of HOM formation of the terpenes and the resulting nucleation and particle growth. While HOM formation was found to be independent of RH, particle formation was clearly pushed back for rising RH. Some speculative explanations for that are presented. From my perspective, very interesting is the new transverse ionization inlet with the curtain gas unit, which could be an alternative to the commonly used Boulder-type inlet, as well as the experimental fact that HOM formation is totally free of water effects. I think that the manuscript meets the criteria for ACP and should be published in this journal. Some minor points should be considered before final acceptance is recommended:*

*1) Line 78-80: The authors mean that the intra-molecular H-shift, or RO2 isomerization, is characterized by a noticeable barrier making this unimolecular step clearly faster with rising temperature. I think it´s not good to say "autoxidation" has a barrier. Autoxidation stands more for the whole process.*

Response: Thanks for the suggestion. We revised the original text accordingly: "The autoxidation of $RO_2$ includes intramolecular hydrogen shifts and $O_2$ additions. Several repetitions of the autoxidation cycle lead to a rapid increase in oxygen content as well as a decrease in saturation vapor pressure. Autoxidation was widely observed in condensed phase reactions, however, it was not considered in the gas phase previously due to the perception of a high barrier for the intramolecular hydrogen shift. This was confirmed by the fact that at higher temperatures, more HOMs are formed than at low temperatures (Frege et al., 2018).

Modeling studies now show that intramolecular hydrogen shifts are fast enough to compete with bimolecular sink reactions (Kurtén , et al., 2015)."

*2) Line 85-86: The Boulder-type nitrate-CIMS by Eisele and Tanner does not suffer from a general problem with water vapour. Only in the case of relatively high RH in the reaction gas water cluster formation during gas expansion can disturb the analysis. Otherwise it works fine. That should be clearly stated at this point. Or have the authors other observations?*

Response: We revised the original text accordingly: "HOM detection by the current commercially available CIMS inlet based on the design of Eisele and Tanner is subject to water cluster influence at high RH (Kürten et al., 2016)."

*3) Line 90: In abstract a RH rangeof 3-90% is stated, and here 0-90%.*

Response: Thanks for pointing this out. We confirm that the RH range in the manuscript is now consistently presented as 3-92%.

*4) Line 114: It would be fine to have a table that compares parameters of the TI inlet with those of the commonly used boulder-type inlet, e.g. reaction times of the IMR, flows, HNO3 concentrations, TIC, detection limits, an estimate of wall losses for RO2s and closed-shell products, etc. Spectra of the same reaction gas recorded with both inlets would provide an impression how good the TI inlet works. Maybe the authors should think about a separate paper describing the TI inlet in detail. Could be important for the community.*

Response: Thanks for the suggestion. We revised the text accordingly: "The calibration factors, defined as $C = [H_2SO_4]/([HSO_4^-]/[NO_3^-])$ (Eisele and Tanner, 1993), for the TI in this position and the commercial inlet were $3.25\times10^{10}$ molecules cm$^{-3}$ and $1.41\times10^{10}$ molecules cm$^{-3}$, respectively. The lower calibration factor for the TI inlet is attributed to the shorter reaction time (~80 ms) compared to the commercial inlet (~200 ms). We note that the reaction time of the TI inlet can be further increased by positioning the ion source assembly further upstream relative to the inlet orifice, which would require a slight modification of the current design. The total ion counts (TIC) of the TI inlet are more than 5 times higher than the commercial inlet, which we attribute to the more direct path of ions through the ion source as well as the use of a Po$^{210}$ radioactive source as compared to the soft X-ray in the commercial nitrate inlet. The limit of detection (LOD) for sulfuric acid, which is defined as three times the standard deviation of the background (Jokinen et al., 2012), is $9.3\times10^4$

molecules cm$^{-3}$ and $1.26\times10^5$ molecules cm$^{-3}$ for the TI and commercial inlets, respectively."
With this revision, we have provided the information necessary to interpret our observations and thus we feel that a table is not necessary. However we will consider the reviewer's suggestion to perform additional characterizations of the TI inlet as well as comparisons to the "Boulder-type" inlet as part of a more detailed, future study.

*5) Line 223: Can the authors derive HOM yields as a result of their experiments?*
Response: An accurate estimate of the HOM yields requires several parameters, many of which must be experimentally determined and are not essential for addressing the main objectives of this study. The transmission efficiency in the TOF vacuum chamber must be determined. While some studies use the transmission curve published in previous literature (Heinritzi et al., 2016), we note that this can change significantly according to voltage settings applied to the TOF mass spectrometer. Another key parameter is the ionization efficiency for each compound, which can vary greatly for the HOM observed in this study: for example, $NO_3^-$ based chemical ionization is reported to have relatively lower sensitivity to the OH-related $RO_2$ radicals compared to $O_3$-related $RO_2$ radicals (Berndt et al., 2016). Since we do not want to compromise our results by providing estimates that are based on many uncertain parameters, we have decided not to include yields in this manuscript and instead focus primarily on mechanism elucidation.

*6) Line 228-229: Reaction of OH-derived RO2 radicals, C10H17Ox, with HO2 leads to H18 products, ROOH.*
Response: The sentence was revised as: "As observed in previous studies, $C_{10}H_{15}O_{6,8,10,12}$ and $C_{10}H_{17}O_{5,7,9,11}$ comprised the $O_3$- and OH-related $RO_2$, respectively (Jokinen et al., 2014). $C_{10}H_{14}O_{5,7,9,11}$ comprised the $O_3$-related closed shell monomers, while $C_{10}H_{16}O_{6,8,10,12}$ and $C_{10}H_{18}O_{6,7}$ comprised the OH-related closed shell monomers (Ehn et al., 2014;Berndt et al., 2016)."

*7) Line 258: Figure S5 should be given in the main body along with Fig.8. From my perspective, total SOA mass is almost water-independent within the experimental errors in the whole RH arrange while particle number drops down by a factor of two.*
Response: We agree that the total SOA mass is almost water-independent. We revised the text related to SOA mass generation accordingly: "SOA mass concentrations remains relatively constant". However, as also stated in the manuscript: "The variability in particle mass concentration as a function of RH for different experiments can be attributed to combined effects of gas phase reactions, condensed phase reactions, physical uptake of water, as well as the re-evaporation of semi-volatile compounds from the wall. We cannot accurately quantify these effects. As a result, although the measured SOA mass concentration remained relatively constant, we cannot draw accurate conclusions from this."

In consideration of article length, we feel that Figure 8 adequately expresses the RH influence on aerosol generation. Figure S5 is more about the details of the experiments, so we think it would be more appropriate to put it in the supplementary information. To clarify the data plotted in Figure 8, we have modified a sentence in the main body to show the size ranges of the generated SOA: "Figure 8 shows the integrated SOA particle number and mass concentrations over the observed diameter range of 10 - 100 nm." Later in that paragraph we state "Peaks in the particle number-size distributions were between 40 and 70 nm (Figure S5)."

*8) Line 319: Here, reagent ion dependent sensitivity for different product classes should be mentioned that could lead to different results for different reagent ions.*

Response: Thanks for the suggestion. The sentence was revised as: "This could be explained by a lower OH/O$_3$ ratio in our experiments, since unlike Berndt et al. we did not provide an extra source of OH to the flow tube. Also, Berndt and coauthors reported lower sensitivity of nitrate reagent ions to OH-related RO$_2$ compared to other reagent ions such as acetate."

**Reviewer #2:**

*The study examines the influences of relative humidity (RH) on the formation of HOM and aerosol from the ozonolysis of three different monoterpenes. HOM have been shown to be central for particle formation over the last years, and therefore knowing their yields under different atmospherically relevant conditions is important. The influence of RH has clearly not been addressed in earlier studies, making this work timely and appropriate.*

*The paper is well-written and the presentation of results is clear and straight-forward. The strength of the paper lies in the finding that HOM yields do not change as a function of RH, and this analysis by itself makes the manuscript worthy of publishing in ACP. The results and subsequent discussion on number and mass formation seem much more speculative. Overall,*

*I have several comments that the authors need to address before potential publication.*

*Major Comments:*
1. *RH control. In section 2.1, the authors describe that the RH in flow tube was achieved by mixing a humidified and a dry flow. The humidified flow was 6.5 LPM while the total in the flow tube was 8.5 LPM. Assuming these flows were at the same temperature (which one would hope) when entering the flow tube, then even if the humidified flow was fully saturated at RH=100%, then the maximum achievable RH should be 6.5/8.5 = 76%. However, the authors state that the RH was probed up to 92%. I do not see how this is possible, unless the humidified flow indeed was several degrees warmer when entering the flow tube. If this were the case, it would not be surprising that particle number formation decreased, as nucleation is an extremely temperature sensitive process.*

Response: As stated in the text, we used a temperature-controlled bubbler to provide the humid air. The first three stages of RH were generated with the bubbler at room temperature, and the fourth stage (85-92%) was generated by heating the bubbler to ~35 °C. The reviewer is correct that the only way to achieve this RH by mixing flows is for the humid air to enter the chamber as a slightly higher temperature than the dry monoterpene flow. Assuming saturation of the humid flow, we estimate that the temperatures of the humid and dry flows were 26 °C and 20 °C, respectively, for the high RH stage only (note: due to the 1 LPM sample flow of the ozone analyzers, the humid and dry flows were actually 6 and 2.5 LPM, respectively). Due to turbulent mixing conditions produced at the inlet end of the flow tube, we expect that the gas temperature quickly reaches the room temperature for the experiments conducted at the highest RH. It is important to note that the first three RH stages were generated at room temperature and the bubbler was heated up only for the highest RH stage, yet particle number concentrations decreased monotonically with each level of RH increase. Nevertheless we cannot discount the influence of temperature in the last RH stage experiments, so we have added the following text: "In order to achieve the highest relative humidity stage, the temperature of the humid flow was saturated at 26 °C before being mixed with room temperature air from the monoterpene source. This resulted in a slightly higher temperature at the inlet of the flow tube, which could contribute to lower nucleation rates (Burkholder et al., 2007). Nevertheless, over the range of the first three humidity stages, up to 65% RH, the gas temperature was constant before, during, and after reaction."

2. *In Figure 6b, when RH is 0, the mass concentration is steadily increasing over the course*

*of nearly one hour. This is very surprising considering that the residence time in the flow tube is 1 minute. Clearly there is some memory effects, and this is not currently discussed at all in the manuscript. Accumulation of semi-volatile on walls is a common phenomenon, and can perhaps explain this behavior. Whatever the reason, it puts very high uncertainties on the results, considering that RH may influence the rate of (re-)evaporation from the walls. Are the measured SOA mass changes really significant when considering this?*

Response: We agree that there may be some memory effects and the semi-volatile species may accumulate and re-evaporate from the wall. The re-evaporation rate could be affected by RH, and if true then this would have some influence on the measured SOA mass concentration. As we mentioned in the paper, this effect is not quantitatively evaluated in our current work. In fact, as we mentioned in the paper, the measured mass concentration likely depends on this effect plus gas phase reactions, gas-particle partitioning, condensed phase reactions, and physical uptake of water. With all these effects and uncertainties, we cannot draw conclusions as to the measured SOA mass concentrations. We have added some sentences in the paper to clarify this: "The variability in particle mass concentration as a function of RH for different experiments can be attributed to combined effects of gas phase reactions, condensed phase reactions, physical uptake of water, as well as the re-evaporation of semi-volatile compounds from the wall. We cannot accurately quantify these effects. As a result, although the measured SOA mass concentration remained relatively constant, we cannot draw conclusions from this observation. In contrast, while particle number concentrations may also be affected by the factors mentioned above, they decreased by a factor of 2~3 with increasing RH."

It is important to note, however, that this phenomenon does not impact our conclusions regarding HOM generation. As shown in section 3.6, almost all the main compounds are ELVOCs or LVOCs, which were not likely to re-evaporate from the wall.

3. *Particle number concentration was in excess of 1e6 cm^-3. At such high concentrations, couldn't nucleation "shut down" at a certain point just because the condensation sink starts to overwhelm the collisions between nucleating agents? Alternatively, the ultimately measured particle number might mainly be limited by coagulation between newly nucleated clusters/particles. While I am not an expert on nucleation, the extreme particle number concentration raise many questions about how far the absolute values can be compared directly, as done by the authors.*

Response: We agree that nucleation can "shut down" at a certain point because the condensation sink starts to overwhelm the collisions between nucleating species. This is the reason we only detected the larger particles (10-100 nm) instead of the nucleation mode particles (<10 nm) at the end of the flow tube. It's very likely that new particle formation only happens at the entrance to the flow tube, while new particles formed at later stages will coagulate with the existing larger particles very quickly. Following this, particle number concentrations decrease due to coagulation and wall loss. The particle coagulation coefficient and wall loss coefficient are simple physical parameters and are well known to be unaffected by $H_2O$. The peak sizes in particle size distributions do not change much under different RH, especially in experiments without OH scavenger (Figure S5). As a result, we assume the particle coagulation term and wall loss term have linear effects on the particle number concentration for different RH. Based on these, the observed decreases in final particle number concentration with increasing RH should reflect the trend in new particle formation rate at different RH. Therefore we feel there is sufficient justification to compare number concentrations at different RH.

We have addressed this issue by confirming the following: (1) while absolute number concentrations are being referred to in the text, the conclusions drawn from the comparisons acknowledge the uncertainties associated with their changes, e.g., stating that concentrations change by factors of "2~3" rather than exact calculated values; (2) we remind the reader that changes in number concentration with RH can also be due to many of the factors that contribute to changes in mass concentration by including the following statement: "In contrast, while particle number concentrations may also be affected by the factors mentioned above, they decreased by a factor of 2~3 with increasing RH."

4. *Related to the precious point, the authors state that since the aerosol surface area is so much lower than the flow tube wall area, the dominant loss term for HOM will be wall loss. This statement seems completely illogical. How are the particles even formed in the flow tube if all ELVOC or LVOC would condense on the walls? The authors need to calculate the actual condensation sink (CS) produced by the particles and I think they will find that the lifetime of HOM due to CS is on the order of seconds, While wall loss will be a slower process. Calculating a coagulation sink in addition might also help address my earlier comment in nr 3.*

Response: We thank the reviewer for pointing out this oversight. Following the reviewer's guidance, we realized that the comparison between the wall surface area and particle surface area cannot represent the relative importance of wall loss rate and condensation rate onto particles. We have calculated the condensation sink (CS) (Kulmala et al., 2001) and wall loss rate (W) (Crump and Seinfeld, 1981;Kürten et al., 2014) using the following equation:

$$CS = 2\pi D \sum_{d_p} \beta_{M,d_p} dp N_{d_p}$$

$$W = \frac{2}{\pi} \cdot \frac{A_{chamber}}{V_{chamber}} \cdot \sqrt{k_e \cdot D}$$

Where, $\beta_{M,d_p}$ is the transition correction factor; D is the diffusion coefficient of $H_2SO_4$, 8.5E-6 $m^2$ $s^{-1}$; $A_{chamber}$ and $V_{chamber}$ are the surface area and volume of the chamber; $k_e$ is the eddy diffusion coefficient, which we use an empirical value of 0.001 $s^{-1}$.

The manuscript was revised as: "The condensation sink (CS) and wall loss rate for a compound with diffusion coefficient of $8.5 \times 10^{-6}$ $m^2$ $s^{-1}$ (e.g., sulfuric acid) were estimated using established methods (Kürten et al., 2014;Crump and Seinfeld, 1981;Kulmala et al., 2001). The calculated CS varied between 0.1-3.5 $s^{-1}$ in different SOA generation experiments, much larger than the wall loss rate (< 0.01 $s^{-1}$). There was about 5-30% variation in CS in each SOA generation experiment from RH=3% to 92%. This amount of variation in CS does not seem to have a noticeable influence on the final concentration of HOMs. To further test the hypothesis that variations in condensation sink do not impact final HOM concentrations, particle free experiments were performed and, again, detected HOM concentrations did not change with RH (Figure 7)."

5. *Section 3.6 discusses volatility estimated of HOM. However, it seems the authors are not aware of the work by Kurtén et al. (2016), entitled "α-Pinene Autoxidation Products May not Have Extremely Low Saturation Vapor Pressures Despite High O:C Ratios". This work certainly needs to be discussed in conjunction with this section, as it raises considerable doubts about the applicability of SIMPOL to vapor pressure estimated of HOM. Using SIMPOL in this work is still completely appropriate, but the large uncertainties should be noted, Also other parts of section 3.6 are questionable, but primarily the authors need to revisit the discussion about oxidation state OSc. The formula they utilize (2\*O:C-H:C) is not applicable in the case that molecules contain (hydro) peroxide functionalities. And according to Table S1, the authors assume this is the case for every molecule under discussion. This will require rewriting all parts where OSc is discussed.*

Response: Thanks for the suggestions. We were aware that the saturation vapor pressure predicted from SIMPOL model may deviate from the real value. There are a lot of uncertainties not only from the method itself but also from simplification methods we used to estimate the functional groups. We revised the paper as "It has to be noted that the group contribution methods very likely underestimate the volatility of the α-pinene autoxidation products due to ignoring intramolecular H-bonding (Kurtén et al., 2016). There may be large uncertainties in SIMPOL method as well as in our functional group estimation process. As a comparison, the Molecular Corridor method (Li et al., 2016), which does not require information on functional groups, was used to estimate the saturation vapor pressure as well."

We agree that the existence of (hydro)peroxide functional groups will cause the $\overline{OS}_C$ to deviate from $\overline{OS}_C$ calculated with this formula *(2\*O:C-H:C)*. The reason is that the oxygen atoms normally have an oxidation state of -2 in most of the functional groups, but have an oxidation state of -1 in (hydro)peroxide groups. Although the deviations from the formula *(2\*O:C-H:C)* was small (within 0.1) in some cases, the (hydro)peroxide may have a bigger influence on $\overline{OS}_C$ if they are present in relatively higher abundance (Kroll et al., 2011), as in our case. We revised the manuscript as: "The average carbon oxidation state ($\overline{OS}c$) was calculated with Equation 3, in replacement of the commonly used formula ($\overline{OS}_C = 2O:C - H:C$), as the second oxygen atom in (hydro)peroxide group does not increase the carbon oxidation state. In equation 3, $n_O, n_C$, and $n_H$ are the oxygen, carbon, and hydrogen numbers in the molecule; $n_{(hydro)peroxide}$ is the number of (hydro)peroxide groups in the molecule.

$$\overline{OS}_C = \frac{2n_O - n_{(hydro)peroxide} - n_H}{n_C} \qquad \text{(Equation 3)}".$$

The values in Table S1 and Figure 9 were changed accordingly.

*Minor Comments*
6. *Section 2.1 describes the flow tube setup, but the description is in many places unclear and ambiguous. For the flows, 0.5 LPM is diluted by 6.5 LPM and then mixed with 2.5 LPM. This adds up to 9.5 LPM. But the total flow was apparently 8.5 LPM? There also seems to be different types of zero air used, although they are named the same. Is some of it bottled synthetic air? Finally, the author should clarify "Gas inlets to the flow tube were made from Teflon tubing that were capped and drilled with small wholes" better, since as written, it remains unclear to me exactly how this part of the setup looked.*

Response: We agree that the description of the flows was a little confusing. The $O_3$ analyzers each require 1 LPM sample air, so this is why the total flow became 8.5 LPM. All of the zero air was generated by the zero air generator (model 737-30, Aadco Instruments). The diffusers consisted of capped ¼" Teflon tubes with ~1 mm holes drilled in the last ~2 cm of the tubes, and is based on the apparatus described and modeled in Ball et al. (1999).

In order to clarify the description we have rewritten the flow tube description as follows:

For these experiments, dry "zero air" was generated with a zero air generator (model 747-30, Aadco Instruments), with NOx and SO2 concentrations each specified to be less than 0.5 ppbv. The monoterpenes were injected into the flow tube using a syringe pump (model NE-300, New Era Pump Systems, Inc.) evaporated into a 2.5 LPM flow of dry zero air. O3 was generated by passing 0.5 LPM dry zero air (79% N2, 21% O2) over a Hg UV lamp (model 90-0004-04, UVP, LLC) and then diluted with 6.5 LPM of humidity-controlled zero air. A temperature-controlled bubbler filled with deionized water was used to generate humid air, and the prescribed RH was achieved by controlling the temperature of the bubbler. An ozone analyzer, described below, sampled at 1 LPM, resulting in a total flow rate of 8.5 LPM and a corresponding reaction time of ~60 s for each experiment. Gas inlets to the flow tube were made from 0.64 cm outside diameter Teflon tubes that were capped and drilled with ~1 mm holes to distribute sample air uniformly into the flow tube, as described and modeled in Ball, et al. (1999).

7. *There is variability in earlier literature about how to write the plural HOM. In some cases also the plural is "HOM", while in other studies they are "HOMs". However, the usage in many places in this work seems questionable. "HOMs dimers", "HOMs volatility", "HOMs production", etc, sound incorrect to me, and HOM should not be in plural form in these cases.*

Response: Thanks. All the "HOMs dimers", "HOMs monomers", "HOMs volatility", "HOMs production", "HOMs products", "HOMs formation", "HOMs generation", "HOMs partitioning", "HOMs concentration", "HOMs spectrum", "HOMs abundance", "HOMs signals" were changed to "HOM …".

8. *In the conclusion, lines 378-383 present various ways through which water could affect HOM formation, but then this is following by a sentence saying that none of them actually take place. Are those lines really needed, or are they just more likely to confuse a reader?*

Response: Line 378-383 is a summary of the possible water-influencing pathways. We feel it is important to introduce these to the reader in order to understand the discussion that systematically analyzes each possible pathway (Line 383-393). Without this introduction, we feel that it would be harder for the reader to understand the context of our analysis.

9. *Line 139, the inlet flow to the mass spectrometer is given as 0,5 LPM, but is it not closer to 0.7 LPM? This is given in some earlier studies, and is closer to the theoretical value for a 0.3 mm orifice.*

Response: The inlet flow was not directly measured, but was subtracted from all the TI inlet flows by all the outlet flows and therefore is subject to measurement uncertainties (which is why we use the approximate symbol "~" in describing the flow). The important point of this section of the text is that the curtain flow exceeded the flow into the mass spectrometer, so we will accept the reviewer's suggestion and replace "~0.5" with the more generally-accepted "~0.7".

10. *Line 304. This is not a very clear sentence, but if I understand correctly, the hydroperoxide should be an alcohol?*

Response: Yes! "hydroperoxide" was revised as "hydroxyl".

11. *Figure 3. The legend says "upstram".*

Response: Thanks, corrected.

12. *Figure 4 need larger axis labels.*

Response: Thanks, revised.

13. *Figure 6c. The legend for the liens is incorrect. O3 radicals are given twice, as are OH monomers. They should be mixed.*

Response: Thanks, corrected.

14. *Figure 9. The molecule in blue in the top-most box: is the keto form more commonly depicted than the enol from given here ?*

Response: As suggested by the reviewer, we checked the literature and found that the aldehyde form is more commonly depicted than the enol form. We changed the related structure in Figure 9 and the volatility prediction accordingly.

15. *There are references to "Kurten at al (2012)" and "Kuerten et al (2016)". As these, to my understanding, are the same person, they should be spelled the same.*

Response: Thanks for the correction. All of "Kürten, Andreas" are uniformed as "Kürten";

All of "Kurtén, Theo" are uniformed as "Kurtén".

*16. While the likelihood of misunderstandings about what particles are studied here is low, the word "aerosol" is not used before in section 2.3. I suggest to include it at least once at a much earlier stage.*

Response: Thanks for the suggestion. The term "SOA" was introduced in the abstract as well as the introduction. The sentence in the abstract is "The ozonolysis of α-pinene, limonene, and $\triangle^3$-carene, with and without OH-scavenger, were carried out under low NOx conditions under a range of RH (from ~3% to ~92%) in a temperature-controlled flow tube to generate secondary organic aerosol (SOA)". We also corrected the first reference to SOA in the introduction (line 85) to "… secondary organic aerosol (SOA)".

**Reviewer #3:**

*The Manuscript describes the RH effects on HOMs formed from oxidation reactions (ozone and OH) of monoterpenes and on the SOA formation. The authors used HRTOF-CIMS to measure gas phase HOMs (monomer and dimers). The main conclusion is that because HOMs are not affected by RH under the present experimental conditions, formation pathways of HOMs may not include water. The review believes this is a reasonable explanation, considering that the autoxidation reactions take place in the condensed phase and at high temperatures, where water is less available. Considering the increasingly important roles of HOMs in NPF and SOA formation, this conclusion is useful to the community.*

*Since the measured particles sizes in this study are between 20-100nm, it would be difficult to discuss the contribution of RH on NPF from these sizes. It is more likely that the precursors of nucleation (or NPF) and 20-100 nm particles have different volatilities and they may be even different precursors. Therefore, the RH effects on these aerosol particles are different from what the community considers regarding to RH effects on NPF (started in Introduction). Rather, the results shown here indicate the effects on SOA formation. So, I would suggest reorganize Introduction and results/discussion to focus on HOMs. Since RH varies in a large range in the atmosphere, it is still important to understand the effects of RH on HOMs formation.*

Response: We agree with the reviewer that it is not correct to associate the formation rate of ~20 nm diameter particles with NPF rates, and we have been careful to avoid such comparisons. We do feel, however, that the observed trends in total particle number do reflect trends in NPF, once we have accounted for effects such as coagulation and wall losses. See our comments to Reviewer 2, Comment 3, above describing how we have addressed such effects. To summarize, we can assume that particle coagulation and wall loss are monotonically related to particle number concentration for different RH (that is, each increases with increased number concentration and neither is greatly affected by RH). Based on this, the observed decreases in final particle number concentration with increasing RH should reflect the trend in new particle formation rate at different RH. Therefore we feel there is sufficient justification to associate changes in overall number concentrations with NPF.

*(1) Line 38-42: The Amazon NPF or lack of it has little to do with RH, rather it is due to high condensation sink or the lack of nucleation precursors.*
Response: As suggested by the reviewer, the Amazon example was removed.

*(2) Line 60: Include Yu, H., (2012). "Effect of amines on the formation of sub-3 nm particles and their subsequent growth." Geophys. Res. Lett. 39: Doi: 10.1029/2011gl050099.*
Response: Thanks, the recommended reference was added.

*(3) Section 2.4. Indicate the exact equation that is used to calculate saturation mass concentrations (C\*). Later, it is stated that the functional groups are also considered, so it would be useful to show the calculation procedure.*
Response: The functional groups used for calculation was summarized in Table S1. As suggested by the reviewer, the calculation equation was added in the caption of Table S1: "The equation used for volatility calculation is $log_{10}P_{L,i}^0(T) = \sum_k v_{k,i} b_k(T)$, where $P_{L,i}^0(T)$ is the liquid vapor pressure of the compound. $v_{k,i}$ is the number of groups of type k, and $b_k(T)$ is the contribution by each group of type k. (Pankow and Asher, 2008)".

*(4) Line 210: What is the source of e5 /cc of sulfuric acid?*
Response: The trace amount of $H_2SO_4$ is generated by OH oxidation of $SO_2$. There is always a trace amount of $SO_2$ in the zero air, and the ozonolysis process will generate some OH radical in the experiments without OH scavenger which will lead to formation of $H_2SO_4$. To clarify this, we have edited the sentence in question as follows: [H2SO4], which arises from the oxidation of trace amounts of SO2 in the aero air, was ~105 molecules cm-3 and was always less than 3% of the most abundant C10 products, suggesting that sulfuric acid plays a negligible role in nucleation and cluster growth in our experiments.

*(5) Line 231: Dimers are more abundant from OH oxidation. This is an interesting result, considering that HOMs formed from ozonolysis are found mostly during the nighttime, where NPF does not take place. So this may explain the importance of dimers on aerosol nucleation.*
Response: In our manuscript, we indicate that $C_{20}H_{32}O_{6-13}$ generation is more abundant in experiments without OH-scavenger, as compared to experiments with OH scavenger. This is because some of the $C_{20}H_{32}O_x$ is generated by the bimolecular reaction between an OH-derived $RO_2$ ($C_{10}H_{15}O_{2n}$) and an $O_3$-derived $RO_2$ ($C_{10}H_{17}O_{2m+1}$). We cannot conclude that dimer formation from OH chemistry is more than from $O_3$ chemistry, because there are other dimers formed, e.g., $C_{20}H_{30}O_x$. Also, because we didn't perform the OH-only chemistry, we cannot compare the yield of OH and $O_3$ chemistry directly.

However, we did find that: "For the HOM products with identical $\overline{OSc}$, OH-derived HOMs have lower volatilities than $O_3$-derived HOMs due to a greater number of (hydro) peroxide groups. As a result, OH chemistry is suspected to be more likely to lead to NPF than $O_3$ chemistry, given the same level of oxidants and VOCs precursors."

*(6) Lines 251-252: Show condensation sink, instead of surface area. The size distribution shows aerosol sizes are 20-100 nm, so condensation actually takes place effectively.*

Response: Thanks for the suggestion. Following the reviewer's guidance and that of Reviewer 2 (see our response to Question 4), we realized that the comparison between the wall surface area and particle surface area cannot represent the relative importance of wall loss rate and condensation (on particles) rate. The manuscript was revised as: "The condensation sink (CS) and wall loss rate for a compound with diffusion coefficient of $8.5 \times 10^{-6}$ $m^2$ $s^{-1}$ (e.g., sulfuric acid) were estimated using established methods (Kürten et al., 2014; Crump and Seinfeld, 1981; Kulmala et al., 2001). The calculated CS varied between 0.1-3.5 $s^{-1}$ in different SOA generation experiments, much larger than the wall loss rate ($\leq$ 0.01 $s^{-1}$). There was about 5-30% variation in CS in each SOA generation experiment from RH=3% to 92%. This amount of variation in CS does not seem to have a noticeable influence on the final concentration of HOMs. To further test the hypothesis that variations in condensation sink do not impact final HOM concentrations, particle free experiments were performed and, again, detected HOM concentrations did not change with RH (Figure 7)."

*(7) Lines 260-261: Indicate yields.*

Response: The sentence was revised as: "The generated SOA particle number and mass concentrations for limonene (2.2-6.0 $\times$ $10^6$ $cm^{-3}$ for number concentrations and 470-1025 µg $m^{-3}$ for mass concentrations) were ~3-12 times greater than for $\triangle^3$-carene (0.3 - 2.0 $\times$ $10^6$

cm$^{-3}$ for number concentrations and 56-86 μg m$^{-3}$ for mass concentrations) and α-pinene (0.4 - 2.2 × 10$^6$ cm$^{-3}$ for number concentrations and 61-130 μg m$^{-3}$ for mass concentrations)."

*(8)Line 306: Decomposition to C5 HOMs. Is this new?*

Response: The detection of C5-C9 HOMs during pinene oxidation is not new. As reported by Ehn and his coauthors (Ehn et al., 2012), $C_5H_6O_7$ is abundant in α-pinene oxidation, as well as some C7 - C9 fragments. However, the formation pathways were unclear. In our manuscript, we just gave a possible formation pathway of C5 - C9 – there may be other fragmentation pathways that we have not considered and that should be the focus of future studies.

**References cited**

[revised manuscript text omitted]